# Effects of relational and instrumental messaging on human perception of rattlesnakes

Erin B. Allison[1], Emily N. Taylor[2], Zackary A. Graham[1], Melissa Amarello[3], Jeffrey J. Smith[3,4], Zachary J. Loughman[1]*

**1** Dept of Organismal Biology, Ecology, and Zoo Science, West Liberty University, West Liberty, WV, United States of America, **2** Biological Sciences Dept, California Polytechnic State University, San Luis Obispo, CA, United States of America, **3** Advocates for Snake Preservation, Silver City, NM, United States of America, **4** The Nature Conservancy, Willcox, Arizona, United States of America

* zloughman@westliberty.edu

## Abstract

We tested the effects of relational and instrumental message strategies on US residents' perception of rattlesnakes—animals that tend to generate feelings of fear, disgust, or hatred but are nevertheless key members of healthy ecosystems. We deployed an online survey to social media users (n = 1,182) to describe perceptions of rattlesnakes and assess the change after viewing a randomly selected relational or instrumental video message. An 8–item, pre–and post–Rattlesnake Perception Test (RPT) evaluated perception variables along emotional, knowledge, and behavioral gradients on a 5–point Likert scale; the eight responses were combined to produce an Aggregate Rattlesnake Perception (ARP) score for each participant. We found that people from Abrahamic religions (i.e., Christianity, Judaism, Islam) and those identifying as female were associated with low initial perceptions of rattlesnakes, whereas agnostics and individuals residing in the Midwest region and in rural residential areas had relatively favorable perceptions. Overall, both videos produced positive changes in rattlesnake perception, although the instrumental video message led to a greater increase in ARP than the relational message. The relational message was associated with significant increases in ARP only among females, agnostics, Baby Boomers (age 57–75), and Generation–Z (age 18–25 to exclude minors). The instrumental video message was associated with significant increases in ARP, and this result varied by religious group. ARP changed less in those reporting prior experience with a venomous snake bite (to them, a friend, or a pet) than in those with no such experience. Our data suggest that relational and instrumental message strategies can improve people's perceptions of unpopular and potentially dangerous wildlife, but their effectiveness may vary by gender, age, religious beliefs, and experience. These results can be used to hone and personalize communication strategies to improve perceptions of unpopular wildlife species.

**Data Availability Statement:** We have deposited our data and code on Dryad: https://datadryad.org/stash/share/X-vR3xg0zGE7Vgo3a-9OWyZhymmYE6DMOWlfItLcK_0

**Funding:** The author(s) received no specific funding for this work.

**Competing interests:** The authors declare no conflict of interest.

## Introduction

Message framing research has been applied across disciplines (e.g., public health, social services, marketing, etc.) and is gaining interest within conservation biology [1–4]. Message framing relates to altering the semantics or articulation of an issue to audiences to evoke a desired response [4, 5]. Specifically, conservation messaging aims to emphasize particular aspects of a subject to influence how people respond to and understand the information in a way that is advantageous for biodiversity [4, 6, 7].

Message framing research is crucial for conservation biology, given that public opinion varies dramatically by taxon [8, 9]. Megafauna such as large carnivores, elephants, primates, and marine mammals hold high symbolic value and are often promoted as flagship species for conservation efforts because the general public favors them, even though many of these species can be dangerous to humans [10–12]. Fundraising campaigns featuring popular species acquire substantial financial aid for conservation; however, these initiatives may be misleading, implying that the need is significant for only these few species.

For example, orangutans are emblematic of threats from oil palm plantations and have successfully garnered attention about this issue, but the public is mostly unaware that palm oil production threatens many other species [13–15]. Plants, invertebrates, and amphibians, for example, are not given the promotion that charismatic fauna receive and therefore earn less conservation support [9, 16–20]. Additionally, studies of unpopular species are often discounted for 'lack of general interest,' requiring added effort to publish findings and secure funding, further hindering research for conservation [21].

Snakes have been objects of phobia and, in some cases, fascination throughout human history because societal biases shape perspectives [22]. Often, snakes are disliked due to cultural myths, stigmas, and deeply rooted fears [12, 23, 24]. Public prejudices of hate, fear, and ignorance of snakes' ecological importance induce anti–conservation behavior in individuals [25–27]. Eid et al. [26] reported that 67% of Jordanian people say they kill snakes they encounter, and over half those with a university degree were unable to distinguish venomous and non–venomous species. Rattlesnakes are among the most persecuted snakes in North America and are often indiscriminately killed [28], a practice that is considered a social norm. While many species of rattlesnakes are found in the United States, few people have seen one in the wild, yet many have negative perceptions about rattlesnakes in general due to misinformation, negative portrayals, and myths passed down through generations. Public events known as rattlesnake roundups are dedicated to culling thousands of wild rattlesnakes each year, in some areas leading to declines in snake populations [*Crotalus adamanteus;* 29, 30]. Roundup perception in many portions of the country has evolved in recent years following changing cultural norms that no longer celebrate the destruction of nature. Several roundups, such as the Claxton Rattlesnake Roundup in Georgia, are now no–kill festivals that celebrate rattlesnakes and promote conservation [31].

Despite widespread and enduring persecution, rattlesnakes provide essential functions to the environment and benefits to humans, including (1) acting as mesopredators in food webs, (2) lowering zoonotic disease transmission risk by eating reservoirs such as rodents, and (3) their venoms represent a resource for current and future inspiration for drug development; [32–35]. The negative reputation of snakes inhibits conservation initiatives and may contribute to their decline [36]. Such extreme antipathy toward snakes generates unique challenges for conservation messaging due to the influence of emotions on individuals' perspectives and behavior.

Historically, science communication was based upon a "knowledge–deficit model," which assumes that humans will adopt a behavior if simply educated about it [37]. But the knowledge–deficit model does not consider how information will be interpreted by individuals [37]. Conservation messaging examines a person's values, attitudes, beliefs, and social norms to

tailor messages and promote behaviors that preserve biodiversity [4, 7, 38]. A significant amount of research over the past decade has been dedicated to understanding how to educate people in ways where they learn and then want to conserve nature. The majority of this work determined that two types of values, instrumental and relational, seem to be the primary drivers for an individual's conservation ethos [39–41].

Instrumental values are derived from a utilitarian foundation, where nature serves humanity and performs a function that aids an individual [42]. An example using snakes would emphasize that preserving snakes leads to reductions in rodent populations, subsequently controlling zoonotic diseases and reducing losses in plant and grain consumption by rodents. Relational values are associated with the philosophical, spiritual, or empathetic ties that individuals have with nature. By conserving nature, and in this example snakes, we make the Earth a more suitable place for humanity and increase our individual experience as part of nature [43, 44]. Using this philosophy, individuals would look at snakes as individuals, and want to conserve them because they are part of nature and nature is worthy of conservation outside a utilitarian narrative.

Relational values also would entail viewing rattlesnakes as individuals with personal strife, challenges, successes, and failures, all parts of the human experience from birth to death. For example, rattlesnakes are dedicated mothers who protect their newborns [45]. Given that humans also dedicate time to raising children, human parents can relate to female rattlesnakes parenting their offspring, which could lead to increased empathy, a pillar of relational and intrinsic value systems in conservation [39–41, 46, 47]. The question then becomes, from a messaging framework perspective, are instrumental or relational value systems applicable to rattlesnake conservation messaging? Optimizing conservation messaging can be a critical component for preserving rattlesnake populations and alleviating their persecution. For example, bats, like snakes, are persecuted due to societal norms, even though they are responsible for controlling insects that serve as vectors of disease and economic loss. Boso et al. [48] found that informing people of the positive ecological impacts of bats led to increased stakeholder engagement and increased acceptance for bat conservation.

Given people's negative view of rattlesnakes, they represent excellent subjects for investigating methods to improve public perceptions of unfavored wildlife among varying demographic groups. If we can change people's minds about rattlesnakes using message framing, techniques garnered herein could be modeled for other wildlife with high conservation need and low public appreciation. In this study, we tested the efficacy of two message framing strategies to promote higher tolerance of rattlesnakes, hoping to shed light on best strategies for improving people's perceptions. Specifically, we investigated:

1) How do socio–demographic factors and prior experience with a venomous snake bite affect participants' baseline perceptions of rattlesnakes?

2) How do the instrumental and relational message strategies change participants' perception of rattlesnakes?

3) How do socio–demographic factors and prior experience with a venomous snake bite influence the impacts of relational or instrumental message strategies on perception of rattlesnakes?

## Materials and methods

We implemented a stratified random sampling method using Google Forms with a survey titled "Rattlesnake Survey." Participants first answered a pre–survey with questions on socio–demographic information and past experiences with venomous snakes (S1 Text), then

completed a pre–Rattlesnake Perception Test (RPT; Table 1), viewed one randomly selected video message strategy, then completed a post–RPT. We divided age (at the time of the survey in 2021) into five generations: Generation–Z (age 18–24 years), Millennials (age 25–40), Generation–X (age 41–56), Baby Boomers (age 57–75), and the Silent Generation (age 76 and older). In the demographic survey, we evaluated the effect of past negative experiences with venomous snakes. Respondents selected yes, no, or not sure to "*you, a friend, or your pet was bitten by a venomous snake.*" Individuals who selected "not sure" to this question were removed from the analysis.

The RPT consisted of six statements and two questions for eight total items that quantify an individual's perception of rattlesnakes (see Table 1). The RPT was similar to tests utilized by Prokop et al. [22] and Pinheiro et al. [49], but surveys in both of these studies were implemented in person and were designed to study specific demographics, while our surveys were taken electronically and our goal was to survey a wide breadth of society. The RPT defines *perception* as a person's emotions for, knowledge of, and behavior toward rattlesnakes, with two survey items reflective of each category following a 5–point Likert scale. We derived an Aggregate Rattlesnake Perception (ARP) score for each participant both pre–and post–RPT by summing the eight items, hereafter called pre–ARP and post–ARP. Negatively phrased statements were scored following a reverse code (Strongly Agree = 1 Agree = 2 Neutral = 3 Disagree = 3 Strongly Disagree = 5). ARP scores ranged from 8 to 40, where low scores (8–18) reflect unfavorable perceptions of rattlesnakes, intermediate ARP scores (19–29) reflect relatively neutral overall perception, and high scores (30–40) illustrate highly favorable perceptions of

**Table 1. Rattlesnake Perception Test (RPT) consisted of 8 items to define perception as emotion, knowledge, and behavior (α = 0.844) following a 5–point Likert scale, where negatively framed items were reverse coded and scored.**

| Category | Survey Item (Statements) | Frame |
|---|---|---|
| Emotion | Rattlesnakes are fascinating animals. | + |
| | If I came across a rattlesnake, I feel that it would harm me. | − |
| Knowledge | Rattlesnakes are generally misunderstood by humans. | + |
| | Rattlesnakes are not important to the ecosystem. | − |
| Behavior | I would like to encounter a rattlesnake in the wild. | + |
| | It is acceptable to injure a rattlesnake, because they could injure me. | − |
| | *Survey Item (Questions)* | *Answer choices* |
| | What is your overall perception of rattlesnakes? | Very Positive |
| | | Positive |
| | | Neutral |
| | | Negative |
| | | Very Negative |
| | Which statement best fits your current perception of rattlesnakes? | I have an affinity towards rattlesnakes. |
| | | I find rattlesnakes interesting. |
| | | I am indifferent towards rattlesnakes. |
| | | I do not find rattlesnakes interesting. |
| | | I have a phobia of rattlesnakes. |

*(5 = Strongly Agree 4 = Agree 3 = Neutral 2 = Disagree 1 = Strongly Disagree)

rattlesnakes. To test for internal reliability and consistency of our survey items, we calculated Cronbach's alpha [50].

After completing the pre–RPT, participants viewed one randomly selected video message (relational or instrumental strategy; details below), then completed a post–RPT. To ensure videos were chosen at random, the survey platform displayed two distinct images in random order, and we asked respondents to select the first image displayed. Each image was linked to the relational or instrumental video message privately hosted on YouTube. The relational video message (90 seconds) focused on rattlesnake social behaviors and presented footage of community dens and birthing rookeries with narration centered on the formation of rattlesnake social bonds, live birth, and parental care (Script available in S3 Text; video available online at https://youtu.be/Bj0EDL4h3SU). We chose anthropomorphic words such as "friends" and "family" to encourage empathy by highlighting traits and behaviors shared with humans. The objective of the relational message strategy was to improve perceptions of rattlesnakes by motivating emotional connections. By contrast, the instrumental video message (120 seconds) emphasized rattlesnakes' contribution to ecosystem services in a literal sense [41]; i.e., encompassing rodent population control, ingesting ticks, and aiding in seed dispersal), with narration explaining how humans may benefit from rattlesnakes, such as reduced disease transmission rates and benefits of controlled rodent populations (Script available in S4 Text; video available online at: https://youtu.be/nZSLAVoV5wA). The objective of this message strategy was to improve perceptions of rattlesnakes by generating practical value for their existence. In both videos, the same female narrator maintained a neutral tone of voice. Notably, the two videos were slightly different lengths and included different footage of snakes.

The survey was dispersed on social media platforms (Facebook, Instagram, Twitter, and Reddit; see S2 Text) from June through November 2021. All individuals who wanted to participate were included. "Rattlesnake Survey" accounts were created on each social media platform for the sole purpose of non–biased dissemination. Using this Facebook account, the survey link was posted on diverse special interest Facebook groups (e.g., gardening, women of impact, local community groups, free items). Most groups were public, though some required permission to join. We joined as individuals and then posted the survey. On Instagram, we used the Rattlesnake Survey account to follow and share the survey with Instagram users that utilized related hashtags (i.e., #TheOnlyGoodSnakeIsADeadSnake, #KillSnakes, #RattlesnakeRoundups, #Snake, #Nature). Because research shows that incentives increase participation [51–53], respondents received the option to enter their email address to win a free iPad. A random number generator selected a winner corresponding to the order of survey participation. The statistical analysis excluded personal identifiers, which were stored separately from survey response data.

### Ethics statement

We received informed consent from each participant in the beginning of the survey. Data from minors (those under age 18) were not included in the study (see below). Each participant had to check a mandatory box on the online survey acknowledging that they read the information within the survey and consented to it. West Liberty University Institutional Review Board approved human subjects research prior to survey cycle initiation (IRB #20210427–1).

### Statistical analysis

We performed all statistical analyses in R (version 3.5.1, R Development Core Team 2019). We first examined our data and removed participants under the age of 18, duplicate or incomplete responses, and non–residents of the United States. Before conducting any statistical analyses,

we visually inspected our data using the performance package in R to assess the normality and collinearity of our model fits. Thus, we determined that all models (described below) did not possess any model fitting issues [54].

**How do socio–demographic factors and prior experience with a venomous snake bite affect participants' perceptions of rattlesnakes prior to viewing a messaging strategy?.** To determine how participants' socio–demographic variables affected their initial perception of rattlesnakes, we used a backward elimination model selection approach to select the socio–demographic variables that best described the data. Specifically, we started with a full model that contained the participants' pre–score as the dependent variable and multiple socio–demographic variables as independent variables. Then, we used the "step" function with the statistical package in R (version 3.5.1, R Development Core Team 2019) to select the best fit model based on the model with the lowest corrected Akaike information criterion (AICc) value at each step [55]. The socio–demographic variables included in the full model were based on the single terms (no interactions) of the participants' gender identity (hereafter referred to as gender), generation, education, residential category (urban, suburban, rural, or remote), religion, and geographic region. Every combination of the full model was run, and the term that most increased the model's AICc value was dropped. This step was repeated until only the best fit model was selected. The best fit model from the backwards model selection procedure contained effects of residential categories, geographic region, generation, religion, and gender. However, only residential category, geographic region, religion, and gender had significant effects, and therefore, a model containing only these variables was used in subsequent analysis as the best fit model. We performed multimodel averaging with this model. We then used the "dredge" function in package MuMIn [56] to fit a set of models with all possible subsets of fixed factors and their interactions (i.e., the full model). After fitting all possible models, we calculated each model's AICc and Akaike weight, the latter variable being the probability that the model best describes the data [55]. We finally calculated each parameter's weighted average, including estimates from all models. The resulting values of parameters were used to calculate the most likely means for each combination of factors.

To determine how prior experience with a snake bite influenced participants' perception of rattlesnakes prior to viewing a messaging strategy, we compared the AICc values of two linear models, both of which had pre–ARP scores as the dependent variable. For the independent variables, one model had the effect of prior experience with a venomous snake (yes = 1, no = 0), whereas a second model had no effect (i.e., a null model). We completed the above analysis using data from all participants, and separately for participants with neutral or negative pre–perceptions of rattlesnakes, which removed the individuals with a pre–ARP score greater than 30 (i.e., the snake lovers).

**How do the relational and instrumental message strategies change participants' perception of rattlesnakes?.** To determine how the relational and instrumental message strategies influenced participants' perception of rattlesnakes, we analyzed data only from participants with pre–ARP < 30 (n = 712 reflecting impartial to unfavorable perceptions of rattlesnakes, because participants that already had positive perceptions are not the target population for these message strategies. For this analysis, we built a linear model with change in ARP score (post–ARP minus pre–ARP) as the dependent variable and message strategy (relational and instrumental) as the independent variable. We then compared this model to a null model using model selection.

**How do socio–demographic factors and prior experience with a venomous snake bite influence the impacts of relational or instrumental message strategies on perception of rattlesnakes?.** For these analyses, we also analyzed data only from participants with pre–ARP scores < 30 (n = 712). We implemented the same backward model selection procedure

described above, separately for participants assigned the instrumental message and the relational message. In all of these models, the dependent variable was the participant's difference in ARP score before and after watching the video messages. For the participants who experienced the instrumental video, backward model selection led to a best fit model with only the religion variable (see Results). Therefore, we did not perform full model averaging in this scenario. However, the backward model selection for participants who viewed the relational message led us to the best fit model with generation, religion, and gender as our independent variables. We then performed the full model averaging procedure described above in this model.

We compared four different models to determine how participants' prior experience with a venomous snake bite influenced the effects of messaging. In all models, the dependent variable was the difference in ARP score after viewing a message strategy. The independent variables in our four separate models were 1) a single independent variable of treatment (relational or instrumental), 2) a single independent variable indicating prior experience with venomous snake bite to the participant, a friend, or a pet (yes or no), 3) an interaction term between these two variables and 4) a null model without any independent variables. The model with the lowest AICc value was determined to be the best fit model.

## Results

1,501 participants responded to the survey. However, 148 people either lived outside of the United States, were minors under the age of 18, or did not complete the RPT and were therefore removed from the dataset. Thus, we analyzed 1,353 individuals' pre–perceptions of rattlesnakes. Participants' self–identified genders were 30% male, 67% female, and 3% non–binary; they identified with the following religious belief systems: 13% agnostic, 17% atheist, 0.6% Buddhist, 48% Christian, 0.2% Hindu, 0.2% Muslim, 9% spiritual, and 10% prefer not to say (PNS). From this sample, 47% of participants had favorable initial perceptions of rattlesnakes (ARP>30) prior to viewing a video message, 10% had unfavorable initial perceptions of rattlesnakes (ARP<18), and 43% had intermediate initial perceptions of rattlesnakes (18<ARP<30). When analyzing the video messages, our sample size was 712 (ARP<30). After this audience viewed a video message, 81% of participants increased their ARP score, 10% decreased their ARP score, and 9% were unaffected. Cronbach's alpha ($\alpha = 0.844$) indicated good internal reliability for the RPT.

Here we report only significant predictor coefficients, followed by model estimates in parentheses, which can be interpreted as either the average ARP score or average change in ARP score (respective to the specific analysis) between that variable and others in its category (i.e., estimate of +2.0 means that the ARP for this variable was 2 points higher than the overall mean).

### How do socio–demographic factors and prior experience with a venomous snake bite affect participants' baseline perceptions of rattlesnakes?

A model containing the participants' gender, religion, residential category, and region best predicted how socio–demographic factors influenced participants' pre–ARP (Table 2). Within the best fit model, the following coefficients were significant predictors of pre–ARP: female, male, agnostic, Christian, Muslim, Jewish, Midwest geographic region, and rural residential category. Table 3 shows how model estimates compare to an intercept represented by agnostic, female, rural residential category, and Midwest geographic region. For example, identifying as Christian was associated with an ARP 3.84 points lower than the overall mean (i.e., intercept), and identifying as male was associated with ARP 2.93 points higher than the intercept.

**Table 2. The most likely model predicting initial Aggregate Rattlesnake Perception scores based on socio–demographic variables (see AICc values).**

| Model | d.f. | ll | AICc | ΔAICc | w |
|---|---|---|---|---|---|
| region X religion X residential category X gender | 17 | –3531.97 | 7098.53 | 0.00 | 0.64 |
| religion X residential category X gender | 13 | –3537.14 | 7100.62 | 2.10 | 0.23 |
| region X religion X gender | 15 | –3535.97 | 7102.39 | 3.87 | 0.09 |
| religion X gender | 11 | –3540.9. | 7104.10 | 5.57 | 0.04 |
| null | 2 | –3632.43 | 7268.87 | 170.34 | 0 |

Regarding how prior experience with a venomous snake related to our participants pre–ARP scores, we found that a model containing the effect of snake bite best predicted our data compared to a null model (ΔAICc = 22.2) and interestingly, individuals who have prior experience with a venomous snake bite viewed snakes in a more positive light, scoring 1.56 points higher than individuals who had no prior experience with a venomous snake bite.

**How do the relational and instrumental message strategies change participants' perception of rattlesnakes?.** When examining how the two video messages influenced a change in ARP for participants with neutral or negative pre–perceptions towards rattlesnakes, a model containing video messaging treatment type best predicted the data. Within the best fit model, both relational (β = –0.77; SEM = 0.28; p = 0.006) and instrumental (β = 4.25; SEM = 0.20; p<0.001) coefficients were strong predictors of difference in score (S1 Table). The relational strategy yielded an average increase of 3.48 points and the instrumental strategy an increase of 4.25 points.

**How do socio–demographic factors and prior experience with a venomous snake bite influence the impacts of relational or instrumental message strategies on perception of rattlesnakes?.** For the relational messaging video, a model containing gender, religion, and

**Table 3. Full model averaging shows the region, religion, residential category, and gender coefficients for initial Aggregate Rattlesnake Perception scores where the intercept is compared to zero and all coefficient estimates are compared to the intercept.**

| Independent variable | estimate | SEM | p |
|---|---|---|---|
| Intercept* (Midwest/agnostic/rural/female) | 28.769 | 0.9168 | < 0.001 |
| Northeast | 1.579 | 1.184 | 0.183 |
| Southeast | 0.336 | 0.603 | 0.578 |
| Southwest | 1.011 | 0.927 | 0.276 |
| West | 1.020 | 0.898 | 0.256 |
| Atheist | 1.263 | 0.732 | 0.085 |
| Buddhist | 0.855 | 2.286 | 0.709 |
| Christian* | –3.837 | 0.647 | < 0.001 |
| Hindu | –0.744 | 3.0483 | 0.827 |
| Muslim* | –11.948 | 3.048 | < 0.001 |
| Jewish* | –3.594 | 1.712 | 0.036 |
| Spiritual | 0.727 | 0.856 | 0.397 |
| Suburban | –0.246 | 0.420 | 0.558 |
| Urban | –0.841 | 0.462 | 0.558 |
| Male* | 2.929 | 0.443 | < 0.001 |
| Non–binary | 1.654 | 1.318 | 0.210 |

*Asterisks indicate statistically significant coefficients that predict the model.

generation best predicted change in ARP (k = 14; ll = –741.74; AICc = 1513.00; ΔAICc = 0; w = 0.80; S2 Table). Within this best fit model, only male (ARP +1.99), female (ARP +3.92), agnostic (ARP +4.11), Baby Boomer (ARP +2.15), and Generation–Z (ARP +6.02) coefficients were strong predictors of change in ARP (Fig 1 and S3 Table). For the instrumental messaging

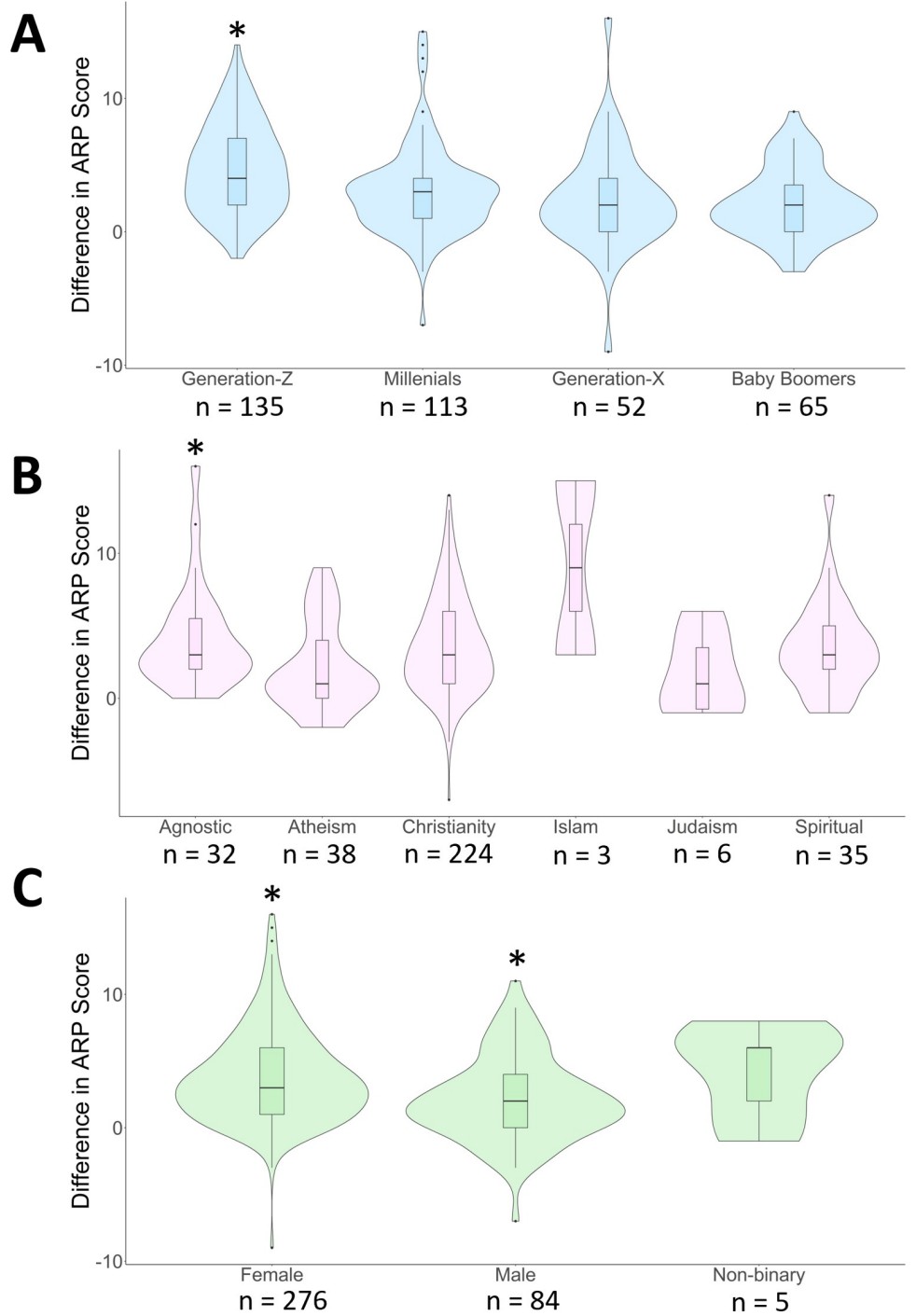

**Fig 1.** The difference in Aggregate Rattlesnake Perception score (post minus pre) for A) generation, B) religion, and C) gender coefficients after viewing a relational message. *Asterisks indicate statistically significant model predictors.

video, a model containing religion best predicted our data. Within the best fit model, agnostic (ARP +3.60), Hindu (ARP +10.50) and Muslim (ARP +9.33) coefficients were strong predictors of change in perception (Fig 2 and S4 Table); however, sample sizes of Hindus and Muslims were small.

After viewing a video message, ARP scores of participants with prior experience of venomous snake bite changed less than those without snake bite experience. A model containing both venomous snake bite experience and video messaging treatment best predicted our data (k = 5; ll = -4196.92–; AICc = 8399.86; ΔAICc = 0; w = 0.80; S5 Table). Within the best fit model, *yes* ($\beta$ = –1.43; SEM = 0.44; p = 0.01) and *no* ($\beta$ = 6.05; SEM = 0.59; p<0.001) coefficients for a venomous snake bite experience strongly explain the variation in change in perception after viewing a video message (Fig 3 and S6 Table).

## Discussion

The empirical literature on the use of conservation messaging to advocate for biodiversity conservation is limited, and current findings are contradictory or unclear [57–59]. To elucidate ambiguous results, recent literature proposes that identification of target audiences and examination of individuals' qualities is key to understanding the effectiveness of conservation

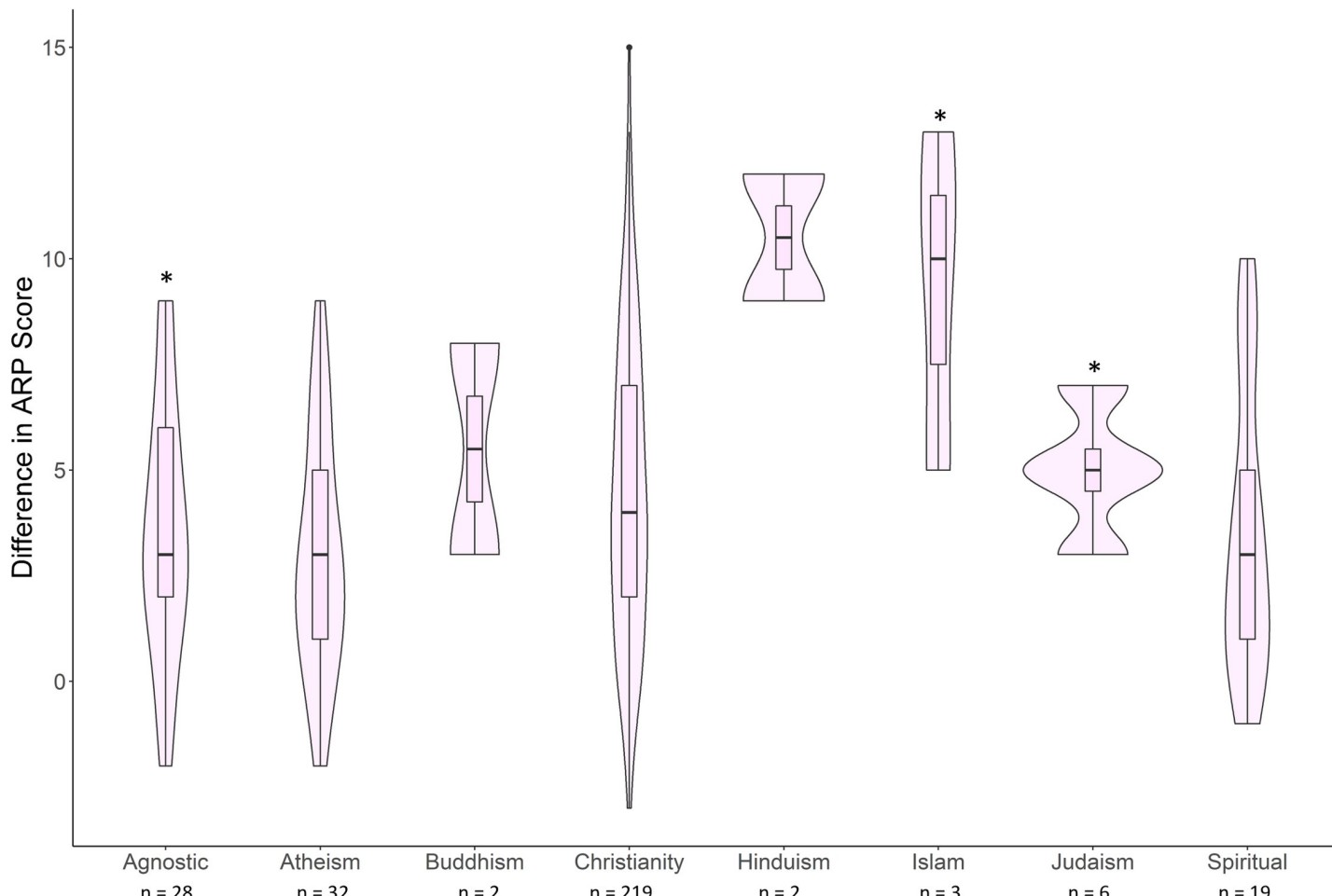

**Fig 2. The difference in Aggregate Rattlesnake Perception score (post minus pre) for religion coefficients after viewing an instrumental message.** *Asterisks indicate statistically significant model predictors.

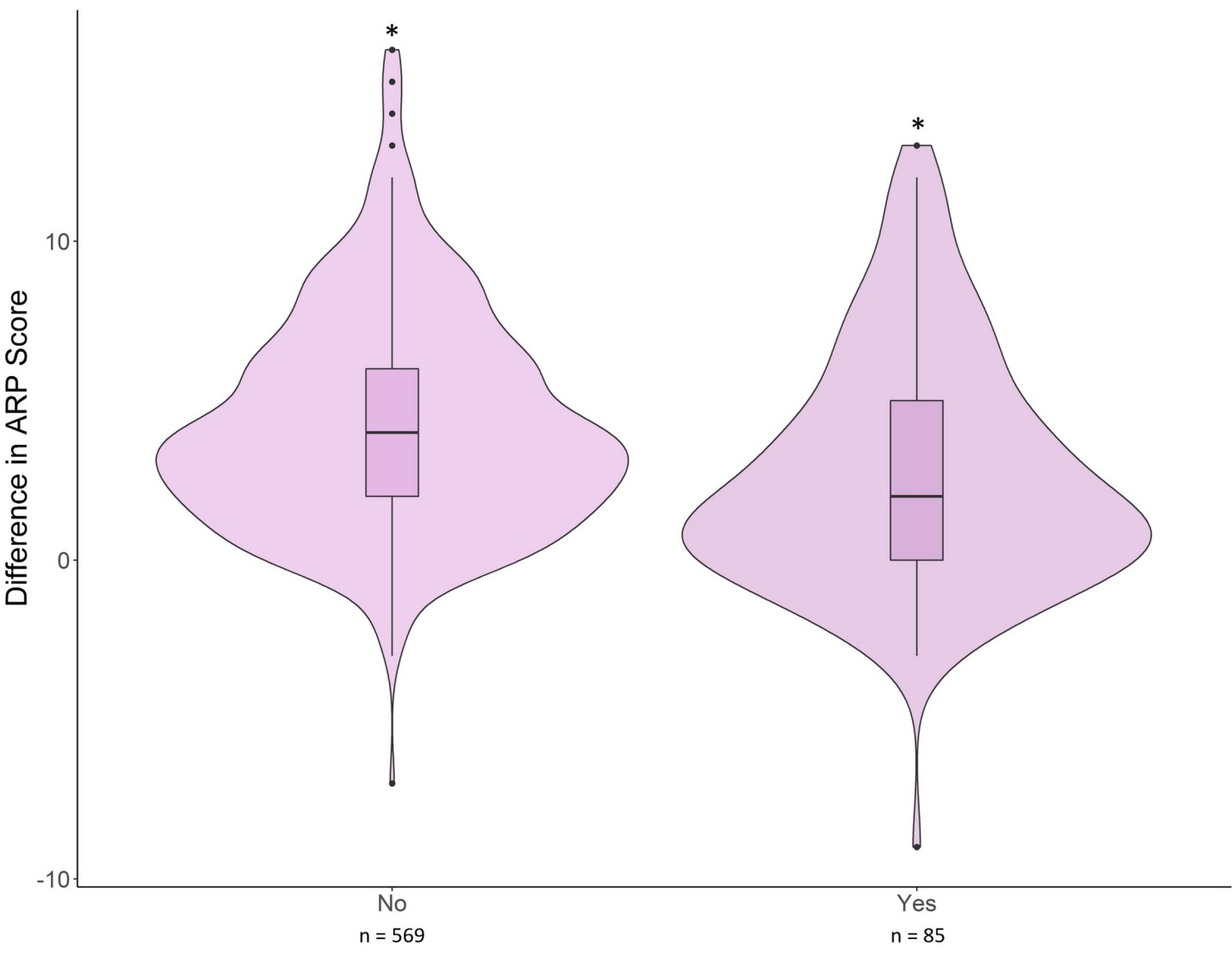

**Fig 3. The difference in Aggregate Rattlesnake Perception (ARP) score (post minus pre) of individuals with or without a venomous snake bite experience (i.e., pet, friend, yourself) after viewing a message strategy.** *Asterisks indicate statistically significant model predictions.

messages [5, 60]. This study aimed to evaluate public perceptions of rattlesnakes with an emphasis on socio–demographics by investigating the efficacy of two message strategies.

'We found that gender, religion, region, and residential category influenced participants' perception of rattlesnakes. The perception of rattlesnakes reported by female–identifying participants was 13% lower than males. This finding is consistent with research showing that females are more likely to have an animal phobia [61] and display greater fear and negative perceptions toward snakes than do males [22, 49, 62–64]. Relatedly, males tend to have a greater interest in less charismatic animals, which may contribute to more favorable perceptions of rattlesnakes than females [65, 66].

Midwesterners and rural residents had relatively positive initial perceptions of rattlesnakes (average ARP = 28). Both the Midwest region and rural residential areas regions are generally known for widespread agricultural practices, so perhaps rattlesnakes are already recognized as beneficial controllers of agricultural pests, thereby producing positive associations in these

residents [67]. Individuals living in these localities may also be more likely to personally experience a harmless encounter with rattlesnakes, thus lessening their risk perception compared to urban areas with lower rattlesnake densities where one's perceptions may be based solely on television and other media sources.

We found that Abrahamic (e.g., Christianity and Judaism) religious affiliation contributes to lower overall initial perceptions of rattlesnakes, and that positive initial perceptions are associated with agnosticism. Scores of those identifying as Christian or Jewish were respectively 14% and 13% lower than the average pre–ARP score. In contrast, the initial perception of agnostics was rather positive (ARP = 31), and they experienced a larger increase after viewing a video message. Their ARP scores were 12% higher than all other categories in this demographic. Given the complex interplay of religion, values, and other variables, we must use caution when attributing difference in perceptions of snakes to religious beliefs [68]. The symbolic and metaphorical potency of animals influence people's perceptions, especially within religious contexts. Positive connotations of snakes are represented in some but not most religions [69]. In Abrahamic religions, snakes often represent wisdom, evil, and are agents of vengeance for God [70, 71]. In the Garden of Eden, the serpent is merely a vessel of Satan, yet the snake itself has been historically demonized for its role in the story [70]. Musah et al. [72] found that 70% of common snake myths in northern Ghana have no scientific basis and are rooted in fear of snakes; these myths are often transferred through religious teachings and contributed greatly to negative perceptions of these organisms [26]. Our study can only provide suggestions about possible relationships between religion and perception of rattlesnakes, so a study aimed at disentangling such relationships might further our understanding of the cultural underpinnings of antipathy towards snakes. Both Muslims and Hindus sample sizes were too small to discuss.

In our study, individuals who had prior experience with a venomous snake bite viewed snakes in a more positive light. While prior research suggests that a painful experience contributes to the fear of animals [26], those with very positive perceptions of rattlesnakes may be more likely to have experiences with a venomous snake bite, due to seeking out and handling venomous snakes, or having friends that do so. We cannot know if the positive perception of rattlesnakes or venomous snake bite experience came first, so we cannot speculate cause and effect, but our findings may indicate that a negative, painful experience does not always increase fear and antipathy toward rattlesnakes, especially if one already has an affinity for them.

Overall, we found that both messaging strategies improved perceptions of rattlesnakes, suggesting that education and positive messaging may improve people's perceptions about unpopular animals. The instrumental message resulted in a slightly greater perceptual increase than the relational video, especially among people identifying as agnostics. The instrumental message was more utilitarian in nature, emphasizing direct benefits (e.g. pest control, disease prevention) of rattlesnakes' presence. In contrast, the relational message of sharing social behaviors of rattlesnakes highlights ethical value with little or no direct profit to people. Although the videos were similar in length and had the same narrator, each video had different content (video footage and script), making it possible that one or more aspects of the content could have contributed to respondents' changes in perception in unintended ways. For example, a respondent might have found the footage showing rattlesnakes eating rodents in the instrumental video to be repulsive. We acknowledge that such content differences could confound our interpretation of the results as purely relational versus instrumental. However, for the most part, the content differences between the videos were explicitly associated with the intended messaging. For example, we showed rattlesnake mothers and babies, and used words like "friends" and "family," in the script of the relational but not the instrumental messaging video as part of an intentional effort to show values with which people might relate.

Broadly, instrumental arguments including ecosystem services can be particularly powerful in illustrating the functional value that certain species provide in maintaining ecosystems [73]. Such value can take shape through economic gain, provisioning of goods and services, environmental regulation (e.g., flood control, water purification, disease regulation, etc.), or supporting environmental processes (e.g., nutrient cycling, soil formation, primary production, etc.; [74, 75]). In the case of rattlesnakes in this study, survey respondents were positively influenced by messages that rattlesnakes help maintain balanced food webs, control rodent populations and the diseases they spread, and otherwise provide benefits to people. Our results suggest that an ecosystem services approach may provoke greater perception shifts due to the importance of advancing human well–being.

While both the relational and instrumental messaging strategies resulted in increased perceptions of rattlesnakes, the relative impacts of each video type varied according to socio–demographic variables, especially religion, age, and gender. People who identified as agnostic experienced notable increases in perception, possibly related to agnostics' open–mindedness and receptiveness to new ideas, suggesting outreach with novel information may improve perceptions of rattlesnakes [74]. Agnosticism is defined as a broad–minded belief in the uncertainty of a god, compared to premodern religions that are associated with historic traditions, communal structures, and values of obedience [68]. Agnostics are not linked to religious beliefs that negatively portray snakes. The belief that unifies agnosticism is a lack of empirical evidence for a god; perhaps merely providing evidence to support rattlesnakes' value may be sufficient to improve perceptions. Whereas the adverse representation of snakes in the Bible likely contributes to Christians' low initial perceptions of rattlesnakes, agnostics may be more progressive in their perspective of snakes and have diverse worldviews.

Generation Z (age 18–24) experienced the highest increase in perception, compared to other generations, after viewing the relational messaging. This suggests that young people are particularly susceptible to changing their perceptions about unpopular wildlife with positive relational messaging interventions. This could be because young individuals tend to be more open–minded and impressionable [76], possibly because they have had less accumulated exposure to negative portrayals of rattlesnakes on television, from friends, family, and other sources. Alternatively, it could be that young adults are more passionate about environmental issues than many older people and are therefore more interested in learning about wildlife–related issues [77].

Baby Boomers (age 57–75) were also particularly influenced by the relational message. Baby Boomers have values closely linked to protection for the welfare of nature [77], and older individuals have increased conscientiousness that is associated with greater empathy [78, 79], so the relational message may have appealed to these participants' feelings of warmth, compassion, and concern for rattlesnakes fueled by the desire to protect nature. In addition, older people have had time to watch the planet change, and it is possible that their observations make them more receptive to messaging that invokes sentimental feelings towards wildlife.

We found that participants identifying as female experienced a greater increase in perception after viewing the relational message than those identifying as males. Importantly, our data showed variation within genders, which is expected given that gender identities do not fall neatly into categories [80] and that a person's susceptibility to relational messaging may be impacted by many more factors than their gender. However, our results did indicate a strong overall impact of gender on change in perception due to relational messaging. The stronger appeal of the relational strategy may relate to a prominent predisposition in females to sympathize with mothers. Herzog Jr et al. [81] found females are socialized from birth to be nurturing, and emotions and feelings are taken into consideration when forming personal morals. Given that rattlesnake parental behaviors were a focal point of the relational video, it is possible that the messaging resonated with females by arousing maternal instincts.

For most people, the perception of rattlesnakes is the result of the accumulation of a life-time's worth of (often negative) messaging about them. Relatively few people have actually seen a rattlesnake in the wild (in our study, only 34% of respondents) and in our study, only one quarter of participants reported that they, a friend, or a pet had been bitten by a rattle-snake. We found that the video messages still led to increases in perception of rattlesnakes among these participants, but the average score increase was 5% lower than that of people with no snake bite experiences. This is likely because personal experience contributes greatly to atti-tude formation, and a painful experience can contribute to the fear of animals [26, 82]. Having this experience is associated with a decreased effect of the video messages, consistent with prior research [83, 84, 85; but see above discussion of participants with snake bite experience and positive pre–perceptions of rattlesnakes].

Larson et al. [86] found that Phoenix area residents felt disgust for rattlesnakes more than fear, and were more likely to have neutral or appreciative beliefs about rattlesnakes if they had interacted with snakes in nature or held snakes at outreach events. Given Larsen et al. [82] and our results, practitioners of snake outreach hoping to change negative snake perspectives should incorporate one on one interaction with snakes and elements of both relational and instrumental messaging for maximum messaging effect. Reliance on relational or instrumental messaging should also be determined based on the demography of the group being educated.

## Conclusions

Message framing aims to accentuate specific aspects of a subject to encourage a particular interpretation and response [5, 87]. Given that different groups of people perceive the world dissimilarly, successful message framing is contingent upon exploring generalizations of socio–demographic groups to customize messages that are impactful to those audiences. Our data suggest that relational and instrumental message strategies are more effective for different demographic groups. Messages incorporating anthropomorphism can be especially effective because the realization of shared features between humans and unpopular wildlife can lead to the development of empathy [88, 89].

While anthropomorphism is recognized as a double–edged sword, it is considered a scien-tifically respectable tool if shared traits are validated and backed by biological evidence [6], and using it as a tool in conservation messaging may be advantageous. Prosocial behavior, intelligence, and the ability to suffer are among the strongest traits for developing relational messaging [90]. We found relational messaging can be more effective for women, young peo-ple (age 18–24), and agnostics through sharing rattlesnakes' pro–social qualities using anthro-pomorphic words. Anthropomorphism can be useful when conducting outreach but must be used critically–grounded in science and knowledge of the natural history and ecology of the species discussed. It is a useful tool to help the public understand the behavior of other species in terms they are familiar with, and can inspire positive conservation action, especially for ani-mals that are historically attributed negative societal norms like snakes.

Deciding whether to use instrumental and/or relational messaging when conducting outreach activities with snakes or other unpopular wildlife should be impacted by considering the demo-graphics and the reactions of the audience. Instrumental messages are particularly useful to assert an anthropocentric rationale that matches attitude functions [91, 92]. For example, ecosystem services messages may motivate attitude shifts when shown to people who value the uses that humans derive from the natural world over the aesthetic or existence of nature itself [91, 93]. Per-sonal values tend to be stable and provide a basis for messaging strategies that can evoke a desired outcome. Understanding value orientations within demographic groups enables practical gener-alizations for the real–world use of conservation messaging. This study identifies general

demographic trends across a broad sample; however, future research investigating message framing on specific target audiences would contribute to science communicators' feasible application of these strategies. It would also be interesting to extend this study to observe whether changes to people's attitudes continue long–term after receiving the message, and importantly whether attitude shifts generate a change in future behavior. Another avenue would be to further investigate those groups in our study for which the messaging had no effect or even *reduced* their ARP. For the latter group, it is possible that watching footage of rattlesnakes, no matter how positive the messaging, made people feel uncomfortable and caused them to regard snakes even more negatively. Studying such effects would help to improve outreach methods even further.

In summary, we found that relational or instrumental message framing can have immediate positive effects on people's perception of one of the most unpopular and reviled animals in North America, but the impact of each message type depended on certain socio–demographic factors, especially gender and religion, but not on other variables like level of education. Consideration of message framing when making outreach and education plans is advantageous for biodiversity conservation because it is an effective technique to influence attitudes, perceptions, and decisions. Altering the general public's perceptions of unpopular taxa can influence pro–environmental attitudes toward maintaining global biodiversity, which may assist in the reduction of extinction rates. Our study found that females are an essential audience to target because their rattlesnake perceptions were initially low but increased significantly after viewing the relational video that showed rattlesnakes engaged in parental care. Additionally, agnostic participants were highly responsive to both types of messaging, suggesting that evidence–based educational interventions may effectively appeal to their worldviews and mitigate potential negative perceptions of rattlesnakes. In contrast, delivering messaging to Abrahamic religious audiences or to individuals with prior experience of snake bite may require more care, as these people have perspectives that are less likely to change. In our study, conservation messaging was an effective strategy to improve the reputation of rattlesnakes, one of the most feared, reviled, and unfairly persecuted animals in North America, which strongly suggests that success could be achieved with other wildlife as well.

## Supporting information

**S1 File. Image of Google Survey used to disseminate Rattlesnake Perception Survey.**
(PDF)

**S1 Text. The demographic questionnaire with items regarding individuals' socio-demographic information and prior experience with rattlesnakes.**
(DOCX)

**S2 Text. Compiled list of information on social media sites used to disseminate the Rattlesnake Survey.**
(DOCX)

**S3 Text. Relational video script.**
(DOCX)

**S4 Text. Instrumental video script.**
(DOCX)

**S1 Table. Treatment coefficient and standard errors of the difference in ARP score (post-pre) reflecting the effect of treatment on perception of rattlesnakes, based on full model averaging.**
(DOCX)

**S2 Table. The most likely model predicting the difference in pre- and post- ARP score for the relational message.**
(DOCX)

**S3 Table. Full model averaging of the effects of generation, religion, and sex on difference in Aggregate Rattlesnake Perception Score (ARP) after viewing the relational message.**
(DOCX)

**S4 Table. Full model averaging of the effects of generation, religion, and sex on difference in Aggregate Rattlesnake Perception Score (ARP) after viewing the instrumental message.**
(DOCX)

**S5 Table. The most likely model predicting the effect of a snake bite experience and difference in Aggregate Rattlesnake (ARP) score.**
(DOCX)

**S6 Table. Full model averaging results for the influence of snake bite experience and treatment on Aggregate Rattlesnake Perception (ARP) score.**
(DOCX)

## Acknowledgments

We would like to thank Advocates for Snake Preservation, J. Short, E. Byrd, and J. Rials for providing rattlesnake video footage. Acknowledgement is due to J. Thompson for developing the videos and A. Sykes for narration.

## Author Contributions

**Conceptualization:** Erin B. Allison, Melissa Amarello, Jeffrey J. Smith, Zachary J. Loughman.

**Data curation:** Erin B. Allison, Zackary A. Graham.

**Formal analysis:** Erin B. Allison, Zackary A. Graham.

**Funding acquisition:** Zachary J. Loughman.

**Investigation:** Erin B. Allison, Emily N. Taylor, Melissa Amarello, Jeffrey J. Smith, Zachary J. Loughman.

**Methodology:** Erin B. Allison, Emily N. Taylor, Zackary A. Graham, Melissa Amarello, Jeffrey J. Smith, Zachary J. Loughman.

**Project administration:** Erin B. Allison, Emily N. Taylor.

**Resources:** Erin B. Allison, Melissa Amarello, Jeffrey J. Smith.

**Supervision:** Zachary J. Loughman.

**Visualization:** Erin B. Allison, Zackary A. Graham.

**Writing – original draft:** Erin B. Allison, Emily N. Taylor, Zackary A. Graham, Melissa Amarello, Jeffrey J. Smith, Zachary J. Loughman.

**Writing – review & editing:** Erin B. Allison, Emily N. Taylor, Zackary A. Graham, Melissa Amarello, Jeffrey J. Smith, Zachary J. Loughman.

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
