## [Decision Letter · Decision Letter 0]

11 Jan 2024

PONE-D-23-40591Effects of relational and instrumental messaging on human perception of rattlesnakesPLOS ONE

Dear Dr. Graham,

Thank you for submitting your manuscript to PLOS ONE. After careful consideration, we feel that it has merit but does not fully meet PLOS ONE’s publication criteria as it currently stands. Therefore, we invite you to submit a revised version of the manuscript that addresses the points raised during the review process.

We look forward to receiving your revised manuscript.

Kind regards,

Anita Mitico Tanaka-Azevedo

Academic Editor

PLOS ONE

Journal Requirements:

4. We note that your Data Availability Statement is currently as follows: "All relevant data and supporting information will be included as online supplemental material."

Reviewers' comments:

Reviewer's Responses to Questions

**Comments to the Author**

1. Is the manuscript technically sound, and do the data support the conclusions?

Reviewer #1: Yes

Reviewer #2: Partly

2. Has the statistical analysis been performed appropriately and rigorously? 

Reviewer #1: Yes

Reviewer #2: Yes

3. Have the authors made all data underlying the findings in their manuscript fully available?

Reviewer #1: Yes

Reviewer #2: Yes

4. Is the manuscript presented in an intelligible fashion and written in standard English?

Reviewer #1: Yes

Reviewer #2: Yes

5. Review Comments to the Author

Reviewer #1: The study is very interesting and can help in the conservation of non-empathetic animals, through simple message framing methods targeted at the population. The comparison of results according to social,gender and age variables are also very newsworthy. The study was carried out in an efficient way, with interesting results and conclusions.

Reviewer #2: The manuscript by Allison et al. aims to test the effects of relational and instrumental message strategies on US residents’ perception of rattlesnakes, through deploying online survey to social media users (n=1,182) to describe perceptions of rattlesnakes and assess the change after viewing a randomly selected relational or instrumental video message. Specifically, authors investigated: 1) How do socio–demographic factors and prior experience with a venomous snake bite affect participants’ baseline perceptions of rattlesnakes? 2) How do the instrumental and relational message strategies change participants’ perception of rattlesnakes? 3) How do socio–demographic factors and prior experience with a venomous snake bite influence the impacts of relational or instrumental message strategies on perception of rattlesnakes?

The study seems to me to be very effective in mapping different people's perceptions of rattlesnakes, and even in measuring the effectiveness of changing these perceptions following new information about the species among residential categories, geographic region, generation, religion, and gender. However, there are some important issues to consider. The main problem observed is the subjective nature of the importance of the relational or instrumental method in the efficiency of changing the perception of the target audience. I find it difficult to disentangle the effect that the videos of each approach have on people's individual preferences. How do we know if the effectiveness of the videos is associated with a particular approach or just with the script, filming, selection of images, etc.? Could there be significant differences between different videos within the same approach, relational or instrumental?

I also believe that the authors could review their decision to consider as "No" those participants who answered "Not sure" to the question "you, a friend, or your pet was bitten by a venomous snake.". It would make more sense to consider this group separately in the analyses, or even to exclude these individuals from the sample, so as not to compromise the reliability of the results by adding uncertainties to the model.

Graham, Z. is listed as corresponding author in PLOSOne system, but Loughman, Z.J. is marked with * on the first page. Please, check who should be the correct correspondent author.

6. PLOS authors have the option to publish the peer review history of their article (what does this mean?). If published, this will include your full peer review and any attached files.

Reviewer #1: No

Reviewer #2: No

---

## [Author Response · Author response to Decision Letter 0]

20 Jan 2024

See attached response to reviewers document.

---

## [Editor Report · Decision Letter 1]

30 Jan 2024

Effects of relational and instrumental messaging on human perception of rattlesnakes

PONE-D-23-40591R1

Dear Dr. Graham,

We’re pleased to inform you that your manuscript has been judged scientifically suitable for publication and will be formally accepted for publication once it meets all outstanding technical requirements.

Kind regards,

Anita Mitico Tanaka-Azevedo

Academic Editor

PLOS ONE
---

## [Editor Report · Acceptance letter]

26 Mar 2024

PONE-D-23-40591R1 

PLOS ONE

Dear Dr. Graham, 

I'm pleased to inform you that your manuscript has been deemed suitable for publication in PLOS ONE. Congratulations! Your manuscript is now being handed over to our production team.

Kind regards, 

on behalf of

Dr. Anita Mitico Tanaka-Azevedo 

Academic Editor

PLOS ONE